# Insight into the Processing, Gelation and Functional Components of Tofu: A Review

Chun-Chi Chen [1,†] [ID], Jung-Feng Hsieh [1,2,†] [ID] and Meng-I Kuo [1,2,*]

1    Ph.D. Program in Nutrition and Food Science, Fu Jen Catholic University, New Taipei City 242, Taiwan
2    Department of Food Science, Fu Jen Catholic University, New Taipei City 242, Taiwan
*    Correspondence: 062998@mail.fju.edu.tw; Tel.: +886-2-2905-2019
†    These authors contributed equally to this work.

**Abstract:** Soybeans are a good source of protein and functional nutrition for humans and are used in the production of many foods, such as soymilk and tofu, which are popular in Asia. Soymilk is a beverage with a milky appearance and is produced by grinding soybeans in water; this raw soymilk is then filtered and heated. Soymilk can be used to produce tofu, a gelatinous food, in a process that generally involves coagulating, forming, and pressing soymilk. Tofu is also a source of functional ingredients, such as isoflavones, that have received much attention for their antioxidant properties. These isoflavones bind to soy protein to form soy protein–isoflavone complexes in tofu. In this review, we examined the processing, gelation and functional components of tofu. First, we introduced novel processing technologies (such as thermal and nonthermal processing) and hydrocolloids that affect the rheological property and texture of tofu. Then, because the coagulation and gelation of soymilk are important steps in the tofu manufacturing process, we also described detailed coagulation and gelation mechanisms of soybean proteins in tofu. Finally, we described the functional compounds and ingredients in tofu. This review provides basic knowledge for the further application of novel processing technology in tofu processing.

**Keywords:** tofu; rheological properties; texture; processing; gelation; isoflavone; functional component

## 1. Introduction

In East Asia, there is a soy product widely and commonly produced from tofu, which is made mainly from soybean. Soy is a precious source of isoflavones, which have received much attention for their anti-oxidant properties, estrogenic activity, and cancer-fighting properties [1,2]. Furthermore, the seed coat of black soybeans is black and contains more anthocyanins and isoflavones than soybeans. Several investigations have mentioned the advantageous bioactive properties of anthocyanins, including antioxidant, anti-obesity, anti-diabetic, and anti-inflammatory activities [3–5].

The traditional method of making tofu is to soak soybeans first, then grind them with water and strain them to make raw soybean milk. Soymilk is boiled and heat-treated for 3-10 min, then cooled to room temperature. Then, it is composed by adding acidic or salt-based coagulants, such as glucono-delta-lactone (GDL) or calcium sulfate for tofu, to heated soymilk [6,7]. Alternative natural coagulants, such as crab shell extract, eggshell, and plant-based extracts, have been proposed by several researchers to add to the health benefits of tofu [8–10]. There are prerequisites for making tofu, which are to dissociate, denature, and aggregate soybean proteins, inhibit microbial growth, reduce the beany odor, and inactivate biological compounds, such as lipoxygenases or trypsin inhibitors, unless performed by thermal treatment [11]. However, in traditional methods, the temperature adjustment of tofu in the production process is time-consuming and laborious. Moreover, there is improper heat treatment affected the quality of tofu. Compared with traditional technologies, ultra-high-pressure homogenization, high-pressure and ultrasonic

non-thermal processing technologies have become alternatives for the food industry to re-innovate high-quality food and reduce processing time and cost [12–14].

The aging population is growing and resulting in increased consumer demand for food products with a variety of textures. Recently, much research has been conducted on approaches to modifying the texture of food products. For tofu preparation, there are some hydrocolloids used as gelling agents or stabilizers, such as carrageenan, chitosan, guar gum, and gum Arabic, to improve texture, water retention, or extend shelf life [15–18]. The addition of hydrocolloids might induce the agglomeration of the soy protein–isoflavone compounds and affect the structure of tofu as well [19]. There is an understanding of the mechanism of interaction among soy protein, hydrocolloids, and functional compounds, which is major in the invention potential for developing new gel textures.

In the food industry, rheology is the study of the deformation and flow of raw material, intermediate products, and final products. Additionally, there are deformation and flow laws in the final product [20]. It is important in understanding textural properties of foods and food processing. Rheological analysis of dynamic oscillation tests was used to observe general structural changes in several food systems under non-destructive conditions [21–23]. This review article discusses the effects of non-thermal processing and hydrocolloids on the texture and rheological properties of tofu, the mechanism of soy protein gelation, and the functional compounds of soybean and tofu.

## 2. Effects of Non-Thermal Processing and Hydrocolloids on the Texture and Rheological Properties of Tofu

### 2.1. Rheology and Texture

The purpose of rheological characterization is to quantify the functional relationship between deformation and stress. There are results in rheological properties, such as viscosity, elasticity, or viscoelasticity. They are primarily concerned with the steady-state flow, oscillatory, and creep measurements for food rheology. The last two methods are usually used for the evaluation of the rheological property of semi-solid foods. They are divided into two sections, oscillatory shear examination, such as small amplitude oscillatory shear, and large amplitude oscillatory shear (SAOS and LAOS). There is a non-destructive rheological technique by SAOS measurement that has been widely used to estimate the viscoelasticity of materials such as strain, temperature, frequency, or a function of time [24]. The determination of shear moduli is allowed by SAOS measurements, including storage or elastic modulus (G′) and loss or viscous modulus (G″). In addition, the phase angle δ and loss tangent tan δ, which is a measure of the ratio of the viscous component to the elastic component, were determined. Moreover, SAOS is restricted to the linear viscoelastic region, where the above shear moduli are not dependent on the magnitude of applied stress or strain and the structure of the tested material is not affected. LAOS measurement describes the rheological properties of food materials at large deformation (i.e., outside the linear zone), closer to real processing and request conditions, chewing and swallowing. A creep test is normally used to evaluate the textural stability of viscoelastic foods. The basic rheological approach has helped researchers study the properties of tofu and the influence of many manufacturing factors. The result is tofu with desirable and consistent texture and rheological properties.

Textural attributes of food products play an important role in consumer appeal, purchasing decisions, and final consumption. Usually, texture assessment is based on the judgment of a sensory panel. The feeding process is a destructive process, including the deformation, flow, cracking, and breaking of food. Texture, however, is the brain's interpretation of the oral sensations of the substance's response and resistance to this deformation [25]. However, texture profile analyzers (TPA) provided a fast and low-cost method to measure texture under well-defined and controlled conditions [26]. TPA is a two-cycle compression test that simulates the first two bites. There are several texture parameters calculated from recorded force versus deformation curves, including hardness, brittleness, cohesiveness, elasticity, cohesiveness, gumminess, and chewiness. The texture

preference data from sensory panels can be related to the rheological data to determine the sensory–rheology relationships using statistical tools [27].

### 2.2. Diverse Technologies for Tofu Making

### 2.2.1. Ultra-High-Pressure Homogenization

There is a constant non-thermal processing technique by ultra-high-pressure homogenization (UHPH). For the food industry, there are potential benefits to producing food of better quality than conventional techniques. The process is based on the same principle as the conventional homogenization method. However, from 100 to 350 MPa, these liquids or substances require colloidal properties to pass through higher-pressure ranges [28]. In the UHPH process, in combination with heat generation, there are concerning factors involved, including cavitation, high shear force, friction, pressure, turbulence, and velocity. The advantages of UHPH have been elucidated by several authors in processing soy products. For instance, in soymilk, the reduction in microbial particle size and burden, denaturation of soy protein and soy flour, formation of more hydrogen bonds in the protein matrix, and preservation of biologically active compounds improve emulsion stability and soy yogurt quality. [29–37].

Huang and Kuo [12] studied the effect of UHPH on the gelation of tofu using the temperature sweep test, followed by the time sweep test under SAOS. Temperature sweep testing involved measuring the dynamic modulus over a temperature range at a constant frequency, constant strain, or stress amplitude. It is essential to investigate phase transitions. They are performed isothermally at constant strain (or stress) amplitude and frequency by time sweep testing. This is useful for monitoring the construction or demolition of buildings. They observed that the gelation temperature of heat-treated soymilk was 76.3 °C and that of 150 MPa UHPH soymilk was lower. Moreover, the gelation temperature of UHPH soybean milk increased significantly with the increase in the number of homogenization cycles. Before gelation, UHPH soymilk observed the association point at the temperature around 41–45 °C, which was not found for heat-treated soymilk. The above results indicated that the gelation behaviors of UHPH-treated soybean milk and heat-treated soybean milk were different because of the different degrees of protein denaturation in soy milk.

Furthermore, Huang and Kuo [12] evaluated the rheological properties of tofu by using a creep test. Among them, there is a suitable explanation for the viscoelastic conduct of tofu through the four-element Burgers model. Compared to those made with UHPH soymilk, tofu made from heated soy milk has higher parameter values of elasticity and viscosity. Furthermore, with the number of homogenization cycles, these parameters of tofu were made and increased from UHPH soymilk.

Li et al. [38] combined HPH heat treatment to make mung bean curd, when the HPH pressure is 0–100 MPa, the hardness of tofu without heat treatment increases. The firmness of tofu after continuous heating was compared with untreated tofu (one-step heat treatment, 95 °C for 5 min). After 10 min of treatment, the tofu with an HPH pressure of 80 MPa had the highest hardness.

UHPH is an advantageous technology for making high-dietary fiber tofu. Liu et al. [30] and Liu and Kuo [39] presented that UHPH reduced the particle size of soybean flour non-thermally denaturing soybean protein and making high-quality tofu. They found that three cycles of 150 MPa UHPH produced tofu that was nearly as hard as heat-treated tofu. The firmness of the tofu suspended with soy flour of approximately 20% was higher than that of the tofu suspended with soy flour by about 15%, which was due to the increased concentration of soybean protein. In addition, they also found that the hardness of the UHPH tofu increased with the circulation and pressure by the researchers. Tofu with different treatments showed similar trends in other texture properties, such as elasticity, stickiness, and chewiness.

### 2.2.2. High-Intensity Ultrasound

Currently, high-intensity (low-frequency) ultrasound (HIUS) technology is attracting considerable attention in the food industry. With specific functional properties, it represents an efficient, environmentally friendly, and reliable alternative to improve food quality or new products in advance [40]. The mechanical waves of HIUS generate the cavitation effect that causes the liquid to quickly form bubbles and rupture, resulting in the formation of strong shear forces and mixing effects. Some researchers have investigated the application of HIUS as preprocessing to facilitate the denaturation and modification of soybean proteins. Additionally, they improved their solubility, emulsifying properties, flow behavior, and gel properties. [41–44].

Lin et al. [13] researched the effect of HIUS treatment made from colored soybeans (black or yellow) on the gelation and the physicochemical property of tofu. During gelation, they observed that the gelation time and rate constant of tofu prepared with heated treatment and HIUS were similar. However, compared to those prepared with HIUS treatment, the tofu arranged with conventional heated treatment had higher $G'$ and $G''$. Moreover, in both the $G'$ and $G''$ values of tofu, there is increased time of HIUS treatment. Some experimental results of tofu demonstrated that the tan δ was between 0.128 and 0.183, which is much smaller than the unity value with the gel structure being generally weak.

Fan et al. [45] evaluated the effects of HIUS treatment at different power settings for okara tofu analogs on the physicochemical characteristics, gelation process, and microstructure. They found that the gelation temperature was the same for all samples, around 80 °C. Compared to the untreated sample, the $G'$ growth rate was higher for the ultrasound-treated okara tofu analog. During gelation, the growth rate first increased and then decreased with the increasing ultrasound power. The TPA parameters of tofu samples were hardness, springiness, gumminess, and chewiness. These parameters increased with increasing the ultrasound power up to 600 W, then they decreased with an increase in ultrasound power. The authors suggested that ultrasound induced the particle size reduction of okara and the protein denaturation, resulting in a dense gel network.

### 2.2.3. High Hydrostatic Pressure

High-pressure processing is also known as ultra-high-pressure processing (UHP) or high hydrostatic pressure processing (HPP). HHP utilizes intense pressure (about 400–600 MPa) at process temperatures of less than 45°C to inactivate harmful pathogens, vegetative spoilage microorganisms, soybean trypsin inhibitors, and lipoxygenase and allows most foods to be preserved with minimal effects on taste, texture, appearance, or nutritional value [46–48]. Zhang et al. [49,50] reported that HHP could modify the structure of soy protein in soymilk and induce tofu gel formation. Moreover, the tofu could be formed with HHP pressure up to 400 MPa, and the gel strength of tofu increased with HHP pressure. Saowapark et al. [14] observed that tofu treated with 400 MPa HHP had higher values of both the $G'$ and $G''$ than those produced by thermal treatment, indicating a stronger structure of HHP tofu.

### 2.3. Hydrocolloids for Tofu Texture Modification

In foods, gelling biopolymers are divided into two main classes, protein and polysaccharides. Understanding the mechanism involved in the interaction between these biopolymers is the main issue to discover their potential in developing a novel gel texture [51–53]. Generally, there are uncharged polysaccharides that maintain protein gel systems by increasing the viscosity of the serum phase [54]. At pH values below the protein's isoelectric point (p*I*), ionic polysaccharides and the positive charges on the protein surface with lower charge densities electrostatically interact [55,56].

Usually extracted from red seaweed, carrageenan is a class of anionic-sulfated polysaccharide. Each repeating disaccharide unit corresponds to the number of sulfate groups, there are three main types, namely, iota (ɩ-), lambda (λ-), and kappa (κ-). Shen and Kuo [17]

demonstrated that on the rheology, water retention, and microstructure, there are effects of various types and concentrations of carrageenan on tofu. They illustrated that the carrageenan-added tofu had higher moisture content and yield than the control tofu. Compared with the control, adding the κ/ι-hybrid carrageenan and κ/ι-mixed carrageenan increased the firmness of tofu, while the addition of K+-κ-carrageenan decreased the hardness of tofu. Some textural properties, such as stickiness, cohesiveness, and chewiness, of different carrageenan tofu also showed similar trends. However, tofu texture decreased with increasing carrageenan concentration. After adding the coagulant, calcium ions act as bridges between the negatively charged groups of the protein aggregates and the carrageenan sulfate groups. Through ionic bonds, $K^+$ simultaneously interacts with two sulfate groups in carrageenan. MacArtain et al. [57] mentioned that a surfeit of counterions leads to a localized aggregation of the chains. Additionally, it causes the formation of gels with thicker networks and lower strength.

Chia seed have mucilage, a hydrophilic heteropolysaccharide, and contain uronic acid [58]. Several researchers investigated the effects of whole chia seed flour (WCSF) on tofu's gelling properties, water binding capacity, texture, and microstructure. Then, compared to control soymilk, WCSF soymilk has a lower gelling temperature [59]. The soybean milk was prepared with different WCSF concentrations and had no significant difference in the rate constants. During gelation, the results indicated that the WCSF added did not affect the gelation rate of soybean milk. The WCSF concentration significantly increased from 0 (control) to 1.5% (*w/w*) and increased the G1 and G2 values of tofu. In the G′ and G″ values, however, there were no significant differences between the control tofu and the tofu samples containing 1 and 2% (*w/w*) WCSF. At higher WCSF concentrations, there was an adverse effect on the formation of the gel network in the gelation of tofu because of more large insoluble particles in WCSF participated.

The γ-polyglutamic acid in the form of H and the γ-polyglutamate in the form of $Na^+$, $K^+$, $NH^{4+}$, $Ca^{2+}$, and $Mg^{2+}$ are non-toxic polypeptides produced by *Bacillus subtitles* through the fermentation process [60]. γ-polyglutamate contains carboxyl groups; in an aqueous solution, there exists a negatively charged polymer. Lee and Kuo [61] found that the addition of γ-polyglutamate increased the gelation temperature and time and decreased the G′ and G″ of tofu. As the molecular weight of γ-polyglutamic acid decreased, the G′ and G″ of tofu decreased significantly. In heated soybean milk, the excess negative charge induced by γ-polyglutamic acid inhibited the coagulation of soluble aggregates. Additionally, in the rheological properties and microstructure, there were changes in tofu.

Chitosan comes is from the deacetylation of chitin and a prepared cationic polysaccharide. Chang et al. [15] observed that a 2% added chitosan increased the gel strength of tofu. This phenomenon was more pronounced for chitosan with higher molecular weight. It might stabilize the tofu structure during entanglement between high molecular weights such as polysaccharides and proteins.

## 3. Gelation of Soybean Proteins during Tofu Processing

Soybean [*Glycine max* (L.) Merr.] is a good source of human nutrition, and a 10% soybean supply is used to produce various food products in the world. In Asia, foods with soybean, such as soymilk and tofu, are popular, reasonably priced sources of high-quality protein [62]. Soymilk is a plant-based beverage made by grinding soybeans in water, and soymilk can be processed into tofu [63]. Therefore, this section describes the process of soy protein gelation during tofu processing, including the denaturation, coagulation, and molding of soy proteins.

### 3.1. Classification and Characterization of Proteins in Soybean

Soybeans are composed of approximately 31% carbohydrates, 35% proteins, 17% fat, 12% moisture, and 5% ash [64]. A total of 90% of soy proteins are salt-soluble globulins, and these soybean proteins contain four protein components with Svedberg coefficients of

approximately 15% 2S proteins, 34% 7S proteins, 42% 11S proteins, and 9% 15S proteins [65]. Several soybean proteins were found in the 2S fraction, including albumin, Bowman-Birk inhibitor, Kunitz inhibitor, Trasylol, and cytochrome C (Table 1). The 7S proteins contain β-conglycinin (7S, β, α, and α' subunits), β-amylase, lipoxygenases, and lectins (hemagglutinins), while the 11S proteins contain glycinin (11S) subunits, including A1a, A1b, A2, A3, A4, A5, B1a, B1b, B2, B3, and B4. 7S and 11S are the two main proteins in soymilk; these two proteins are composed of several protein subunits. 7S is a trimer composed of three subunits, namely, β (42–53 kDa), α (57–76 kDa), and α' (57–83 kDa), which are assembled by hydrogen bonds and hydrophobic forces. These trimers can consist of the following different subunits: ααα, ααα', αββ, ααβ, αα'β or α'ββ [66,67]. 11S is a hexamer with a molecular weight of approximately 320 to 360 kDa. Each of the subunits (58–69 kDa) can dissociate into basic (B; 18–20 kDa) and acidic (A; 31–45 kDa) polypeptide chains, which are linked by disulfide bonds [68]. A total of five 11S subunits have been named, including G1 (A1aB2), G2 (A1bB1b), G3 (A2B1a), G4 (A3B4), and G5 (A5A4B3) [64]. In addition, the 15S proteins are composed of polymers of soybean proteins.

**Table 1.** The protein composition included 2S, 7S, 11S, and 15S in soybean.

| Svedberg Coefficients | Protein Name | Molecular Weight (kDa) | Isoelectric Point | References |
|---|---|---|---|---|
| | Albumin | 12.0 | - | [69] |
| **2S** | Bowman–Birk inhibitor | 7.8 | - | [64] |
| | Kunitz inhibitor | 21.5 | - | [64] |
| | Trasylol | 8-21.5 | 4.5 | [65] |
| | Cytochrome C | 12 | 10.2–10.8 | [65] |
| | β-Conglycinin, α' subunit | 72 | 5.5 | [70] |
| **7S** | β-Conglycinin, α subunit | 70 | 5.0 | [70] |
| | β-Conglycinin, β subunit | 50 | 5.8 | [70] |
| | β-Amylase | 61.7 | 5.0–6.5 | [65] |
| | Lipoxygenases | 102.0 | 5.7–6.4 | [65] |
| | Lectins (hemagglutinins) | 102.0 | - | [65] |
| | Glycinin, A1a subunit | 37.0 | 5.4 | [70] |
| | Glycinin, A1b subunit | 37.0 | 5.2 | [70] |
| | Glycinin, A2 subunit | 37.0 | 5.0 | [70] |
| | Glycinin, A3 subunit | 42.0 | 5.1 | [70] |
| | Glycinin, A4 subunit | 36.0 | 4.8 | [70] |
| **11S** | Glycinin, A5 subunit | 10.0 | - | [71] |
| | Glycinin, B1a subunit | 20.0 | 7.2 | [72] |
| | Glycinin, B1b subunit | 20.0 | 8.2 | [72] |
| | Glycinin, B2 subunit | 18.0 | - | [73] |
| | Glycinin, B3 subunit | 21.0 | 9.3 | [72] |
| | Glycinin, B4 subunit | 20.2 | - | [73] |
| **15S** | protein polymers | 600 | - | [65] |

### 3.2. Denaturation of Soybean Proteins in Soymilk

Soymilk is a plant-based beverage made by grinding soybeans with water and heating raw soymilk. The denaturation processes used in tofu production included heating treatment, ultrasonic treatment, microwave treatment, and ultra-high-pressure homogenization (Table 2). The denaturation processes used during soymilk processing cause soybean proteins to denature and dissociate. These processes cause proteins to form and unfold protein filaments due to hydrophobic interactions and disulfide bond formation [63]. Under the heating treatment conditions, soymilk was heated to denature soybean proteins at 80 °C for 30 min, 95 ± 3 °C for 10 min, 80 °C for 70 min, or 80 °C for 70 min [61,74–76]. During the heating treatment processing, the natural soybean proteins in raw soymilk were denatured through heat to form denatured soybean proteins. Hsiao et al. [77] reported

that compared to unheated soymilk samples, the surface hydrophobicity of heated soymilk samples was significantly higher. The increase in $H_0$ is due to the greater availability of the hydrophobic region after the aggregate dissociates. Heating soymilk caused the unfolding and pattern protein filaments for the 11S and 7S proteins, resulting in a loss of hydrophilicity and a negative charge on the surface. After ultrasonic treatment, microwave treatment, and ultra-high-pressure homogenization processing, the natural soybean proteins in raw soymilk were also denatured to form denatured soybean proteins. Soymilk was ultrasonically treated with an ultrasonic cell disruptor/homogenizer at an amplitude of 30% and a frequency of 20 kHz for 30 min to denature soybean proteins, while the soymilk was heated in a domestic microwave oven at 210 W for 90 s for protein denaturation [13,78]. Moreover, soymilk was homogenized in a valve-mode homogenizer to denature soybean proteins [79].

**Table 2.** Processes of denaturing soybean proteins in raw soymilk.

| Methods | Key Process | References |
|---|---|---|
| | Soymilk was heated at 80 °C for 30 min. | [61] |
| **Heating treatment** | Soymilk was heated at 95 ± 3 °C for 10 min. | [74] |
| | Soymilk was heated at 80 °C for 70 min. | [75] |
| | Soymilk was heated at 85 °C for 30 min. | [77] |
| **Ultrasonic treatment** | Soymilk was treated with ultrasonic at 50 °C for 30 min. | [13] |
| **Microwave treatment** | Soymilk was heated in a microwave oven at 210 W for 90 s for protein denaturation. | [78] |
| **Ultra-high-pressure homogenization treatment** | Soymilk was homogenized in a homogenizer at the pressure of 150 MPa for protein denaturation. | [12] |

*3.3. Coagulation of Denatured Soybean Proteins Using Coagulants*

As shown in Table 3, metal ions, polysaccharides, organic acids, and enzymes were used as coagulants to produce tofu. Metal ions, such as calcium chloride, magnesium chloride, and bittern, can precipitate soybean proteins [70,79,80]. Most of the 11S acidic (A1a, A1b, A2, A3, and A4), 11S basic (B1a), and 7S (α′, α, and β) proteins in soymilk were coagulated by 5 mM calcium chloride, while the 5 mM magnesium chloride coagulated the 11S subunit, 7S subunit, Bd 30K, sucrose-binding protein 2, β-amylase, trypsin A inhibitor and lectins in the soymilk. Polysaccharides, such as chitosan and propylene glycol alginate (PGA), also precipitated soybean proteins. It was found that most soybean proteins, including 11S and 7S (α′, α and β), were observed when 0.5% chitosan was added, while 7S (α′, α and β), 11S acidic subunits and 11S A3 and 11S basic protein were precipitated by 0.9% PGA in soymilk [77,81]. Furthermore, the addition of organic acids, including glucono-delta-lactone, citric acid, and malic acid, lowered the pH of soymilk [82–84]. When the pH of soymilk drops to approximately 5.8, soybean protein aggregation occurs through the acidification of soymilk [85]. When the pH of soy proteins approaches their isoelectric points, random aggregation of neutralizing proteins occurs through hydrophobic interactions and hydrogen bonding [85]. Furthermore, the added 4 mM glucono -δ-lactone (GDL) induced the aggregation of 11S basic, 11S acidic, and 7S (α′, α, and β subunits) proteins [82]. Finally, 11S basic (B1a, B1b, and B3), 11S acidic (A1a, A1b, A2, A3, and A4), and 7S proteins in soymilk were polymerized by 1.0 units/mL of transglutaminase [71].

**Table 3.** Coagulation of denatured soybean proteins in soymilk using coagulants.

| Component | Coagulants | Key Process | References |
|---|---|---|---|
| **Metal ion** | Calcium chloride | Crosslinking of soybean proteins by the protein-$Ca^{2+}$-protein bridges. | [79] |
| | Magnesium chloride | Cross-linking of soybean protein by magnesium ions. | [70] |
| | Bittern (nigari) | Cross-linking of soybean proteins with metal ions. | [80] |
| **Polysaccharide** | Chitosan | Cross-linking of soybean proteins with the positively charged amine groups in chitosan. | [81] |
| | Propylene glycol alginate (PGA) | Hydrophobic association between PGA nonpolar soy protein side chains and ester groups. | [77] |
| **Organic acid** | GDL Citric acid Malic acid | The pH value approaches the isoelectric point of soybean proteins, leading to protein aggregation. | [82] [83] [83] |
| **Enzyme** | Transglutaminase (TGase) | TGase catalyzes the cross-linking reaction between soybean proteins. | [72] |

*3.4. Molding of Coagulated Soybean Proteins for Tofu Production*

In soymilk, as mentioned above, coagulation occurs due to the cross-linking of protein molecules with coagulants. After the coagulation process, the coagulated soybean proteins must be molded and the whey must be removed to prepare firm tofu [86]. As shown in Table 4, several processes of molding coagulated soybean proteins in soymilk were investigated. For example, Joo and Cavender [80] used a metal ion (nigari) as a coagulant to produce tofu. The tofu curds were poured into the mold with cheesecloth, and then a 1 kg weight was added to the cheesecloth for 2 h [81]. Cao et al. [83] used organic acids, including GDL, citric acid, and malic acid, as coagulants. The tofu curds were immediately transferred to a mold (7 × 7 × −7 cm) lined with cheesecloth and used an in-house press to remove the whey for 30 min at an 8 g/cm$^2$ press [84]. Furthermore, No and Meyers [18] used polysaccharide (chitosan) as a coagulant to prepare tofu. There were gently transferred to a container (10.5 × 7.5 × 10.0 cm) lined with a single layer of cheesecloth from tofu curds. Then, press for 15 min using bricks (3.8 kg). Finally, Zhu et al. [76] reported that the enzyme (TGase)-containing tofu curds was immediately transferred to a plastic mold and pressed for 3 h. This section describes tofu processing from raw materials to processing conditions in the research progress.

**Table 4.** The processes of molding coagulated soybean proteins in soymilk.

| Coagulants | Molding Process | References |
|---|---|---|
| **Metal ion (nigari)** | The tofu curd was pressed for 2 h using bricks (1 kg) to remove the whey. | [80] |
| **Organic acid (GDL, citric acid and malic acid)** | The tofu curds were pressed for 30 min at 8 g/cm$^2$ to remove the whey. | [83] |
| **Polysaccharide (chitosan)** | The tofu curd was pressed for 15 min using bricks (3.8 kg) to remove the whey. | [18] |
| **Enzyme (TGase)** | The tofu curd was transferred to a plastic mold and pressed for 3 h. | [76] |

## 4. Functional Compounds and Ingredients from Soybean to Tofu

This review provides an overview of the new knowledge of the bioactive components of soy and tofu. Soybeans are mainly used in the preparation of many foods of which tofu is the most popular in Asia [87].

Soybean (*Glycine max*) is a soybean plant whose seeds are abounding in protein. In Asia, the soybean is a native plant with oval and spherical fruits [88]. Due to different strains, the color of the seed coat is yellow, light green, brown, and black; they are also known as soy, green, and black beans. In addition, there is an abundance of small amounts of non-nutrients in soybeans and soy products with potential health benefits; commonly, in different studies, called phytochemicals [89]. They contain many functional compositions, such as isoflavones, anthocyanins, phytosterols, phytic acids, phenols, saponins bioactive proteins, and peptides from soybeans. Their many ingredients are considered antinutrients in traditional nutritional theory [90,91]. In the past, it was discovered that they might exert beneficial and therapeutic effects on health. The purpose of this paragraph gives prominence to the knowledge about the bioactive compositions of soybeans or soy products, especially isoflavones and anthocyanins. Soybean is an important nutritional and human health crop that originated in Asia. Around the world, soy is used in healthy diets due to its high content of isoflavones and anthocyanins.

### 4.1. The Bioactive Components of Isoflavones in Soybeans

Isoflavones in soybean or soy products such as tofu, are free aglycones and conjugates. Recently, there have been many studies that demonstrated twelve isoflavones consisting of four chemical forms, each of which contains compounds, such as acetyl-β-glucoside, malonyl-β-glucoside, and aglycone in soybeans. They found genistein, daidzein, and glycidin, which are in the three major groups of isoflavones from soybeans. Isoflavones are classified as aglycones, including genistein, daidzein, and daidzein; β-glycosides include daidzin, genistin, and glycitin; acetylglycosides include 6″-O-acetyldaidzin, 6″-O-acetylgenistin, and 6″-O-acetylglycitin, and malonylglycosides include 6″-O-malonyldaidzin, 6″-O-malonylgenistin, and 6″-O-malonylglycitin flavin [92]. Many studies have illustrated that isoflavones from soybeans decreased cholesterol levels, thus reducing the risk of cardiovascular disease, and inhibiting cell proliferation. In addition, they have anti-cancer, anti-aging, and anti-inflammatory properties, as shown in Table 5.

**Table 5.** Major health effects brought about by isoflavones.

| Soybean Content | Health Effects | References |
|---|---|---|
| **Isoflavones** | Decreased cholesterol levels. | [93,94] |
| | Reducing the risk of cardiovascular disease. | [95] |
| | Inhibiting cell proliferation. | [96] |
| | Anti-cancer. | [97] |
| | Anti-aging. | [98] |
| | Anti-inflammatory. | [99] |

### 4.2. The Bioactive Components of Anthocyanins in Soybeans

In recent years, anthocyanins have been proven to have health-promoting properties for the human body. Anthocyanins are flavonoid polyphenols, which are common in our daily diets, particularly in yellow, red, black, purple, or blue cereals such as soybean [100]. Anthocyanins mainly exist in the shape of heterosides in nature. The aglycone modus of anthocyanins, also known as anthocyanins, is structurally based on the flaviliumion or 2-phenylbenzopyran, consisting of hydroxyl and methoxy groups in various positions [101]. More than 635 anthocyanins have been identified, based on the number and position of the hydroxyl and methoxyl moieties [102]. Anthocyanin-rich extracts from soybeans have extensive effects on human medical treatment, such as anti-oxidant, anti-diabetic, anti-

obesity, and anti-inflammation activity, as well as the prevention of Alzheimer's disease and cardiovascular disease compounds, as shown in Table 6.

**Table 6.** Major health effects brought about by anthocyanins.

| Soybean Content | Health Effects | References |
|---|---|---|
| Anthocyanins | Anti-oxidant. | [103–105] |
| | Anti-obesity. | [106–108] |
| | Anti-diabetic. | [109–112] |
| | Anti-inflammation activity. | [113–116] |
| | Prevention of Alzheimer's disease. | [117,118] |
| | Prevention of cardiovascular disease. | [119,120] |

*4.3. The Bioactive Components of Isoflavones in Soybean Products*

In East Asia, soybeans have always been treated as an important source of protein. However, in the Western world, there is an increased interest in and consumption of soy products because of knowledge of the nutritional and functional properties of soybeans. People are looking for more nutritious and healthy products, focusing on the results of people's food choices and lifestyles [121].

In the general food industry, soy products, also known as tofu, are often used as desserts and side dishes. The production of tofu usually goes through the processes of soybean screening, soaking, grinding, filtering, boiling, coagulation, pressing, preservation, and packaging [122,123]. However, after a series of treatments in soy products, some nutrients are also affected by different processing treatments, such as protein, isoflavones, or anthocyanins [124]. Therefore, some researchers have used different processing methods to form tofu, a soft cheese curdled from heated fresh soy milk, to which calcium or magnesium salts have been added to maintain the good protein in tofu. This increased the shelf life of foods such as tofu, which is an excellent source of protein and isoflavones and stored under ambient conditions for up to 1 year [125,126]. In addition, researchers used lactic acid bacteria to ferment tofu. The results showed that the bioavailability of isoflavones in tofu after fermentation was higher than that of conventional tofu. The bioavailability of isoflavones may be affected by their chemical forms and heating processes of the food industry's manufacturing processes [127]. In tofu, both genistein and daidzein mostly exist in the form of glycosides. Then, after using tofu, isoflavones are hydrolyzed by gut microbes into unconjugated forms, namely daidzein, genistein, and glycidyl glycosides, which are estrogenic and bioavailable [128]. In addition, some research has shown that malonyl genistein is the greatest isoflavone form in soybeans, followed by malonyl daidzein, daidzein, and genistein. After using tofu, in the intestine, isoflavones are extensively metabolized; they are absorbed or transported to the liver and go through enterohepatic circulation. In the intestine, the bacterial glucosidases break down the sugar and release the biologically active isoflavones, soy zein, and genistein. Furthermore, bacteria-active isoflavones are bio-transformed into specific metabolites. [128,129].

*4.4. The Bioactive Components of Anthocyanins in Soybean Products*

Anthocyanins, a type of plant polyphenol, have received increasing attention in recent years, mainly due to their potential health benefits and applications as functional food ingredients. Soaking beans is the first step in the tofu-making process. Traditionally, however, soybeans have been soaked in water for long periods, resulting in the loss of nutrients in soybeans, such as anthocyanins [130–132]. Therefore, some researchers have studied the effect of soaking time on the bioactive components in soybeans, and the results have shown that long soaking times lead to the loss of nutrients in soybeans [133,134]. In addition, during tofu production, different processing conditions affected bioactive components, especially heat treatment conditions. Several researchers have investigated the effect of heat treatment conditions on bioactive components in soybean products. The results demonstrated that different processing techniques caused complex changes in the

chemical composition, and heat treatment led to the degradation of anthocyanins and the release of conjugated components [135,136]. The differences in the changes of anthocyanins were induced by heat treatment in soybean due to the different distribution and content of individual phenolic compounds in the seed coat and cotyledon [137,138].

## 5. Conclusions

Soybeans have many health benefits, such as hypoglycemic, hypolipidemic, anti-obesity, and anti-cancer properties, due to phytonutrients such as bioactive peptides, saponins, phytosterols, and phenolic compounds. As a result of these nutritional and health benefits of soybeans, soybean products are gaining popularity worldwide. Tofu is a semi-solid soy protein product and is part of the prescriptive cuisine of East Asia. Recently, there have been several novel ingredients and processing conditions for tofu, such as coagulant, non-thermal processing (ultra-high-pressure homogenization, high-intensity ultrasound, and high hydrostatic pressure), and hydrocolloids would improve the rheological property and texture of tofu and relieve anti-nutrients such as oxalate, phytic acid, and saponins, thereby enhancing the functions of soybeans in the food industry. Soy protein isolates, soy protein concentrates, soy textured proteins, soy flour or grits, and traditional tofu products are made from soybean, which is not only nutritious but also therapeutic. In the future, long-term soy product studies are needed to further demonstrate the processing, gelation, and functional components of tofu.

**Author Contributions:** Conceptualization, C.-C.C., J.-F.H. and M.-I.K.; writing—original draft preparation, C.-C.C., J.-F.H. and M.-I.K.; writing—review and editing, C.-C.C.; project administration, J.-F.H. and M.-I.K. All authors have read and agreed to the published version of the manuscript.

**Funding:** This research received no external funding.

**Institutional Review Board Statement:** Not applicable.

**Informed Consent Statement:** Not applicable.

**Conflicts of Interest:** The authors declare no conflict of interest.

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
