# Peer review of "Insight into the Processing, Gelation and Functional Components of Tofu: A Review"

_processes, doi:10.3390/pr11010202_

Round 1

Reviewer 1 Report

Overall Comment on the review:

In the Review Article entitled Insight into the processing, gelation and functional components of tofu: A review authors presented a comprehensive literature review of basic knowledge for the application of novel processing technology in tofu processing. Overall, I enjoyed reading this Review Article, which was written clearly and conscientiously. According to that, this review can be recommended for publication in its present form in such a reputable scientific journal.

Author Response

We sincerely appreciate your time spent reviewing this manuscript.

Reviewer 2 Report

This paper contributes to the understanding and analysis of recent developments in the properties and processing technology of tofu. The review of rheological behaviour is relevant not only to establish quality control, but also to contribute to the understanding of the principles of flow behaviour that allow the study of the "structuring" of this material and its relationship with texture, sensory perception and stability, as well as the processing of this food. I find the review offered by the manuscript useful in this context. At this respect  I would like to include just a few comments

             Line 59.  Please correct typographical error:

The authors stated: Rheology is the study of the deformation and flow of raw materials, intermediate products and final products in the food industry [20].

In fact, Rheology is the study of the deformation and flow of matter, which can be applied to any product, including polymers, gels,.. and similar materials. But, the  correct  definition in [20] reffers to : Food rheology is "the study of the deformation and flow of raw materials, intermediate products and final products in the food industry" [White, G. W. 1970. Rheology in food research. J. Food Technol. 5, 1-32]

             2.1. Rheology and texture: Please consider the following:

The characterization in food rheology includes different tests that can be grouped into tensile, compression or shear tests, the latter being one of the most used techniques. As indicated in the manuscript,  the shear application can be performed in oscillatory, continuous flow, creep, stress relaxation,...

       The main difference between the two mentioned oscillatory test methods is that SAOS is restricted to the linear viscoelastic region  where the rheological  response is not  dependent on the magnitude of applied stress or strain and  the structure of the tested material is not affected, while the LAOS  region is  outside the linear zone,  and as stated by the authors, the rheological behavior can be related to many processes which require very large deformations such as chewing and swallowing, or other processing conditions.

Author Response

(The authors gave the same response as above.)

Reviewer 3 Report

This review is an example of a simple non-informative review written to increase the number of articles of authors. Most information is simple, resembling copy and paste of abstracts and common knowledge information. No good insights, future trends, or advantages of new technologies are provided.

1. Section 2.1. is very simple. Although important for the review, it brings no revision of the literature. It is nested under non-thermal processing, but the section does not mention anything regarding non-thermal processing. The authors should rethink the structure of the manuscript sections.

2. Section 2.2. resembles a copy and paste of abstracts than a real review. The authors give the main results for some published articles, but there are no comments on the advantages or disadvantages of the processes. No input on best practices was made.

3. Section 3.3. This section starts with a repetition of information already provided in the previous sections.

4. Sections 3 and 4 are very simple and provide no new information.  

Author Response

(The authors gave the same response as above.)
